# Genomic Characterization of *Escherichia coli* Isolates from Alpaca Crias (*Vicugna pacos*) in the Peruvian Highlands: Insights into Functional Diversity and Pathogenicity

**DOI:** 10.3390/microorganisms13071533

**Published:** 2025-06-30

**Authors:** Celso Zapata, Lila Rodríguez, Yolanda Romero, Pedro Coila, Renán Dilton Hañari-Quispe, Oscar Oros, Victor Zanabria, Carlos Quilcate, Diórman Rojas, Juancarlos Cruz, Narda Ortiz, Richard Estrada

**Affiliations:** 1Facultad de Medicina Veterinaria y Zootecnia, Universidad Nacional del Altiplano de Puno, Puno 21001, Peru; pcoila@unap.edu.pe (P.C.); rhanari@unap.edu.pe (R.D.H.-Q.); odoros@unap.edu.pe (O.O.); vzanabria@unap.edu.pe (V.Z.); 2Instituto de Investigación en Bioinformática y Bioestadística, Lima 15024, Peru; limrp.bioinfo@gmail.com (L.R.); yolanda.bioinfo@gmail.com (Y.R.); 3Dirección de Desarrollo Tecnológico Agrario, Instituto Nacional de Innovación Agraria (INIA), Lima 15024, Peru; ceqp2374@yahoo.com (C.Q.); diormanr@gmail.com (D.R.); 4Estación Experimental La Molina, Dirección de Supervisión y Monitoreo en las Estaciones Experimentales Agrarias, Instituto Nacional de Innovación Agraria (INIA), La Molina, Lima l2791, Peru; jcruz@inia.gob.pe (J.C.); nortiz@inia.gob.pe (N.O.)

**Keywords:** *E. coli* genetic diversity, antibiotic resistance in alpacas, virulence factors in alpaca pathogens

## Abstract

Diarrhea in alpaca crias significantly impacts livestock health in high-altitude regions, with *Escherichia coli* as a common pathogen. This study analyzed 10 *E. coli* isolates from diarrheic and healthy alpacas using whole-genome sequencing to assess genetic diversity, virulence factors, and antibiotic resistance. Predominant sequence types (ST73, ST29), serotypes (O22:H1, O109:H11), and phylogroups (B2, B1, A) were identified. Virulence profiling revealed ExPEC-like and EPEC pathotypes, while resistance genes for β-lactams (*blaEC-15*), fosfomycin (*glpT_E448K*), and colistin (*pmrB*) were prevalent. These findings highlight the need for genomic surveillance and antimicrobial stewardship to manage *E. coli* infections in alpacas and reduce public health risks.

## 1. Introduction

Diarrhea in neonatal alpacas (crias) represents a significant health and economic concern for livestock production, particularly in high-altitude regions where alpacas are primarily raised. Infectious agents, including bacterial pathogens such as *Escherichia coli* (*E. coli*), are common contributors to gastrointestinal disorders in these animals. *E. coli*, a versatile and ubiquitous bacterium, displays a broad range of pathogenic behaviors, from commensalism to the development of severe enteric diseases, depending on its virulence repertoire and host susceptibility [1,2].

Pathogenic *E. coli* is classified into pathotypes based on virulence factors, such as Enteropathogenic *E. coli* (EPEC) and Extraintestinal Pathogenic *E. coli* (ExPEC), which cause localized and systemic infections, respectively. The ability to acquire resistance to antibiotics and environmental stressors further complicates the treatment and management of *E. coli* infections. These characteristics necessitate robust molecular and genomic analyses to understand the pathogen’s epidemiological and genetic diversity [3,4].

Alpacas are economically and culturally significant in regions like the Andean plateau of Peru, where they are valued for their fiber, meat, and milk. However, there is limited information about the molecular and genomic characteristics of *E. coli* in alpacas, particularly their resistance and virulence profiles. Understanding these parameters is critical for devising effective intervention strategies and ensuring sustainable livestock production [5,6].

This study aimed to isolate and characterize *E. coli* strains from alpaca crias with diarrhea and healthy controls to explore their genetic diversity, antibiotic resistance, virulence factors, and phylogenetic relationships. By leveraging whole-genome sequencing, serotyping, and bioinformatic tools, we provide a comprehensive analysis of *E. coli* isolates, shedding light on their role in alpaca health and potential public health implications [1,7].

## 2. Materials and Methods

### 2.1. Sampling

Fecal samples were obtained from eight alpaca crias showing symptoms of diarrhea, along with two samples collected from healthy individuals. All alpacas were maintained under the same living conditions (Appendix A). The crias were selected based on fecal residues extending from the perianal region to the hock. Before sampling, the perianal area was cleaned using cotton soaked in warm boiled water. A sterile swab was then inserted with rotational movements and placed into a test tube containing 1 mL of 1% peptone water. The sample tube was labeled and stored in a cooler with refrigerant gel to maintain refrigeration until transported to the Microbiology Laboratory at the Faculty of Veterinary Medicine and Animal Science, National University of the Altiplano. The samples were delivered within 3 h of collection.

### 2.2. Isolation and Biochemical and Molecular Characterization

The bacteriological workflow for conventional isolation and identification followed the guidelines [8]. For the primary culture to isolate *E. coli*, the streaking technique on MacConkey agar was used, and plates were incubated under aerobic conditions at 37 °C for 24 h. Presumptive identification was based on the appearance of colonies on MacConkey agar, which were circular and convex with well-defined edges and exhibited a pink coloration with a finely mottled surface. For definitive identification, the following conventional biochemical tests were performed [9]: inoculation on Triple Sugar Iron Agar (TSI), Simmons Citrate Agar, Urea Broth, and Lysine Decarboxylase Broth. Genomic DNA was extracted using the Wizard^®^ Genomic DNA Purification Kit (Promega Corporation, Madison, WI, USA) following the manufacturer’s instructions. The quality of the extracted DNA was assessed by measuring its concentration with a NanoDrop ND-1000 spectrophotometer (NanoDrop Technologies, Wilmington, DE, USA) and verified through 1% agarose gel electrophoresis, ensuring the reliability of the samples. The near-full 16S rRNA gene was then PCR-amplified using primers 27F (AGAGTTTGATCMTGGCTCAG) and 1492R (GGTTACCTTGTTACGACTT), followed by Sanger sequencing of the amplified products. All tests were duplicated, with three biological replicates per strain included in each experiment. The isolates were identified as *E. coli* by comparing their sequences using the NCBI BLAST tool 2.16.0 (https://blast.ncbi.nlm.nih.gov/, accessed on 8 March 2025).

### 2.3. DNA Sequencing, Assembly, and Genome Annotation

The Illumina TruSeq Nano DNA Library Prep kit was used for preparing the DNA library, and the sequencing was performed on the Illumina NovaSeq 6000 sequencing system (Iowa City, IA, USA) with paired-end reads of an average length of 150 nucleotides. Raw reads were processed for quality control and adapter trimming using FastQC v0.11.8 and TrimGalore, followed by assembly with isolateSPAdes v3.15.2. Contigs smaller than 200 bp were excluded from the final assemblies. Genome annotations were conducted using the NCBI Prokaryotic Genome Annotation Pipeline (PGAP) v5.2. The genomic identities were subsequently verified through taxonomic classification based on the Genome Taxonomy Database, utilizing GTDB-Tk to classify v6.1.0. The Illumina sequencing genomes have been deposited in the GenBank database of the NCBI and are publicly available under the following accession numbers: JBJFLZ000000000, JBJFMA000000000, JBJFMB000000000, JBJFMC000000000, JBJFMD000000000, JBJFME000000000, JBJFMF000000000, JBJFMG000000000, JBJFMH000000000, and JBJFMI000000000.

### 2.4. Multi-Locus Sequence Typing (MLST), Serotype, and Phylogroup Determination

Multi-locus sequence typing (MLST) was performed using the mlst 2.23.0 (https://github.com/tseemann/mlst, accessed on 10 January 2025), applying the *Escherichia*-specific scheme to determine sequence types (STs). Serotypes were identified using ECTyper 2.0.0 [10] with default parameters, allowing precise classification based on antigenic markers. Phylogroups were assigned using EzClermont 0.7.0 [10]. These methods provided critical insights into the genetic and epidemiological characteristics of the isolates.

### 2.5. Antibiotic Resistance Gene and Virulence Factors and Pathotypes Identification

Antibiotic resistance genes were identified using abritAMR 1.0.19 [11], which employs the AMRFinderPlus database, and ResFinder 4.1 [12] from the Center for Genomic Epidemiology (CGE), which also relies on a curated database of known resistance determinants. Functional classifications of the identified genes were performed using the Resistance Gene Identifier (RGI) tool from the Comprehensive Antibiotic Resistance Database (CARD) [3]. This analysis allowed the categorization of resistance genes by mechanism of action and their potential impact on antibiotic efficacy. Virulence genes were identified using VirulenceFinder 2.0 from CGE [1,13,14] and ABRicate 1.0.1 (https://github.com/tseemann/abricate, accessed on 10 January 2025) with the Virulence Factor Database [15], which facilitated the detection of genes associated with pathogenicity. Pathotypes were assigned by comparing the virulence gene profiles with the classification criteria used by Enterobase for *E. coli*. Three binary heatmaps annotated with metadata such as host, geographic origin, MLST and Serotype were generated using ggtree 3.14.0 [16]: one showing the identified antibiotic resistance genes and mutations and two showing virulence genes—one including all identified genes and another highlighting key genes selected based on the literature.

### 2.6. Plasmid Replicon Identification and Genomic Island Prediction

Genomic islands were predicted using IslandViewer 4 [17], which integrates the IslandPick, IslandPath-DIMOB, and SIGI-HMM methods. Gene annotations from the integrated predictions were extracted, and previously identified resistance and virulence genes were cross-referenced with these annotations to determine their presence within genomic islands. Plasmid replicons were detected using PlasmidFinder 2.1 [5] from the Center for Genomic Epidemiology, with thresholds of 90% coverage and 95% nucleotide identity. Plasmid contigs were annotated using Prokka v1.13.4 [18] to identify genes with potential roles in antibiotic resistance or virulence, which were subsequently cataloged to assess their contribution to the isolates’ pathogenic potential.

### 2.7. Pan-Genome Analysis

Pan-genome analysis was performed using Panaroo 1.5.2 [19] (strict clean mode, MAFFT aligner, -a core, core threshold: 99%) to characterize core and accessory genomes across the ten *E. coli* isolates and compare gene content between diarrheagenic (DEC) from Puno-Lampa and Puno-Melgar. Invalid genes were removed. The resulting core genome alignment was used to infer a maximum likelihood phylogenomic tree of the 10 alpaca isolates in IQ-TREE 2.4.0 [20] (ModelFinder, 1000 bootstraps). A gene presence/absence matrix was visualized as a heatmap alongside the phylogenomic tree using the ggtree 3.14.0 [16], with genes clustered by Manhattan distance and Ward’s method to identify region- and cluster-specific gene groups. Principal Component Analysis (PCA) of the accessory genome (excluding core genes) was performed with prcomp in base R 4.3.3 to evaluate patterns associated with geographic origin, health status, phylogroup, MLST, and serotype.

### 2.8. Functional Annotation and Enrichment Analysis

Genes from the pan-genome (core and accessory), as identified by Panaroo 1.5.2 [19], were annotated using EggNOG-mapper 2.1.12 [21] to assign Gene Ontology (GO), KEGG Orthology (KO), and KEGG Pathway annotations. GO enrichment analysis was first performed on the core and accessory genomes to provide an overview of functional processes in each genomic compartment. To explore functional differences between groups, targeted GO, KO, and KEGG Pathway enrichment analyses were conducted using genes unique to each group of diarrheagenic *E. coli* (DEC) isolates—UNAP1–4 (Puno-Lampa) and UNAP7–10 (Puno-Melgar). Enrichment analyses were performed using the enricher function from the ClusterProfiler 4.14.0 [22] package. Only genes with relevant GO, KO, or KEGG Pathway annotations were included.

### 2.9. Phylogenomic Analysis

To investigate the evolutionary relationships and genomic diversity of the *E. coli* isolates from alpacas, we conducted a phylogenomic analysis. For a broader comparative context, we retrieved all 469 *E. coli* genomes from Peru available in EnteroBase [2,23], encompassing isolates from humans, swine, chickens, and alpacas. The 10 alpaca genomes were also uploaded to EnteroBase for joint clustering analysis. Core genome Multi-Locus Sequence Typing (cgMLST) was performed using the EnteroBase cgMLST V1 scheme, which profiles allelic variation across 2513 conserved genes. Genomes were then grouped into hierarchical clusters (HierCC) based on cgMLST similarity. In *E. coli*, HierCC defines clusters at 11 levels based on cgMLST allelic distances; the HC1100 level corresponds to cgST Complexes, grouping genomes that differ by fewer than 1100 core alleles [24]. A Minimum Spanning Tree (MST) was constructed using the RapidNJ algorithm and visualized in GrapeTree [25] to explore genetic relatedness across isolates, and was colored by Host source, Phylogroup, Pathovars, and HC1100 clusters representing cgST Complexes.

Based on the MST, 70 genomes representing the closest genetic neighbors to the alpaca isolates were selected for phylogenomic analysis. Selection was based on visual proximity to alpaca isolates in the MST, including those assigned to the same cgST Complexes (HC1100 level). Two additional genomes from alpacas in Peru and two from bovines in the USA were retrieved from NCBI to enhance the representation of livestock-associated isolates. The selected genomes were processed with Panaroo 1.5.2 [19] to generate a core genome alignment, which was used to infer a maximum likelihood phylogenomic tree in IQ-TREE 2.4.0 [20]. ModelFinder [26] was used by IQ-TREE to identify the best-fit substitution model, and branch support was assessed with 1000 bootstrap replicates. The resulting tree was visualized and annotated with metadata such as host, geographic origin, and pathovar assignment using ggtree 3.14.0 [16].

## 3. Results

### 3.1. MLST, Core Genome MLST (cgMLST), Serotype, and Phylogroup

MLST analysis revealed that the majority of isolates belonged to sequence types ST73 (*n* = 4) and ST29 (*n* = 3), with one isolate identified as ST342 and two that could not be assigned to any known ST (Table 1). All ST73 and ST29 isolates were from diarrheic alpacas (Appendix A). The two untyped isolates (UNAP5 and UNAP6), recovered from healthy alpacas, exhibited complete allelic profiles, but their combinations did not match any known ST in the Achtman MLST database (Appendix A). Their profiles represent potentially novel combinations not currently catalogued in the MLST database. Core genome MLST (cgMLST) analysis further differentiated isolates into core genome sequence types (cgSTs), demonstrating genetic variability among the 10 genomes not captured by traditional MLST (Table 1).

Serotyping revealed five distinct O:H serotypes among the isolates, with O22:H1 and O109:H11 being the most prevalent. All O22:H1 isolates belonged to phylogroup B2 and ST73 and were obtained from diarrheic alpacas in Puno-Lampa. Similarly, O109:H11 was associated with phylogroup B1, ST29, and was found in isolates from diarrheic alpacas in Puno-Melgar. Phylogroup analysis showed that most isolates from diarrheic alpacas belonged to phylogroups B2 and B1, with only one assigned to phylogroup A. The only isolates from healthy alpacas (both from Puno-Lampa) were assigned to phylogroups A and B1 (Table 1 and Appendix A).

### 3.2. Phylogenetic Relationships Between the 10 E. coli Isolates

A phylogenetic analysis was conducted to assess the genetic relationships among the ten *E. coli* isolates from alpacas in Puno, Peru. The phylogenetic tree revealed two major clusters, which largely corresponded with phylogroup classifications, MLST types, and serotypes (Figure 1). Cluster 1 comprised UNAP1–4, i.e., all isolated from diarrheic alpacas in Puno-Lampa. These isolates exhibited high genetic similarity, forming a tightly clustered group. All belonged to phylogroup B2, MLST 73, and serotype O22:H1. Cluster 2 was more diverse, containing isolates from phylogroups A and B1, originating from both Puno-Lampa and Puno-Melgar. Within this clade, UNAP7, UNAP8, and UNAP10 (diarrheic from Puno-Melgar) clustered with UNAP5 (healthy from Puno-Lampa), forming a subclade within phylogroup B1. UNAP7, UNAP8, and UNAP10 shared MLST 29 and serotype O109:H11, while UNAP5 had an untyped MLST and serotype O8:H2. A separate subclade within Cluster 2 included UNAP6 (healthy from Puno-Lampa) and UNAP9 (diarrheic from Puno-Melgar), with both belonging to phylogroup A. Despite their phylogenetic relatedness, these isolates differed in MLST classification (UNAP9: MLST 342, UNAP6: untyped MLST) and serotype (UNAP6: O7:H25, UNAP9: O145:H25).

The phylogenetic clustering generally corresponded with phylogroups and MLST types, with isolates from the same phylogroup typically clustering together. However, some subclades included isolates from different geographic regions and health statuses (such as UNAP6 and UNAP9), indicating variability within the population. These phylogenetic relationships provide a framework for examining resistance and virulence profiles among the isolates.

### 3.3. Phenotypic Antimicrobial Susceptibility Testing (AST)

All 10 *E. coli* isolates underwent antimicrobial susceptibility testing (AST) to assess their resistance profiles. All isolates were phenotypically susceptible to ampicillin, nitrofurazone, tetracycline, enrofloxacin, ciprofloxacin, nalidixic acid, and piperacillin-tazobactam. No resistance was observed against any of these antibiotics under the conditions tested.

### 3.4. Antimicrobial Resistance Profile in E. coli Isolates

Analysis of antimicrobial resistance genes and mutations identified distinct resistance profiles that largely corresponded with the phylogenetic clustering previously described (Figure 2). Cluster 1 (UNAP1–4, diarrheic alpacas from Puno-Lampa) exhibited an identical resistance profile, with no observed variation. This was consistent with their close genetic relationships, as they also shared strain characteristics (phylogroup B2, MLST 73, and serotype O22:H1).

Cluster 2 displayed more variation in resistance profiles. UNAP7, UNAP8, UNAP10 (sick from Puno-Melgar), and UNAP5 (healthy from Puno-Lampa) shared a common resistance profile, reflecting their phylogenetic grouping. Within Cluster 2, UNAP6 (healthy from Puno-Lampa) and UNAP9 (sick from Puno-Melgar) displayed slight differences in resistance profiles. Both carried *emrD* instead of *acrF* and *mdtM* for efflux pumps, but UNAP9 harbored additional resistance determinants (*fosA7.5* and *nfsB_W94STOP*), which were absent in UNAP6.

### 3.5. Genotypic and Phenotypic Resistance Comparison

A comparison between the predicted resistance phenotypes based on genomic analysis and the phenotypic resistance observed in AST revealed discrepancies (Figure 2). Despite the presence of the antimicrobial resistance genes and mutations mentioned, all isolates were phenotypically susceptible to fluoroquinolones, β-lactams, and nitrofuran-class antibiotics. The expected resistance to fluoroquinolones based on the detection of *acrF* and *emrD* was not observed, as all isolates remained susceptible to ciprofloxacin, enrofloxacin, and nalidixic acid. Similarly, although *blaEC* and *blaEC-15* were detected (Figure 2), all isolates were susceptible to ampicillin, suggesting a lack of β-lactam resistance under the conditions tested. The *nfsB_W94STOP* mutation, which is associated with nitrofurantoin resistance, did not correlate with phenotypic resistance to nitrofurazone. The antibiogram did not include fosfomycin or colistin, preventing verification of the predicted resistance to these antibiotics.

Analysis of resistance genes revealed a broad spectrum of antibiotic resistance mechanisms (Figure 2). Efflux pump-associated genes such as *acrF*, *emrD*, and *mdtM* were widespread, indicating multidrug resistance. Specific mutations in *glpT* (e.g., *glpT_E448K*) were linked to fosfomycin resistance, while *blaEC* and its variants (*blaEC-15*) conferred resistance to β-lactams. Mutations in *pmrB* were associated with colistin resistance, and a nonsense mutation in *nfsB* was linked to nitrofurantoin resistance in certain isolates.

### 3.6. Virulence Profile in E. coli Isolates

A total of 197 virulence-associated genes were identified among the *E. coli* genomes isolated from 10 alpacas (Appendix A). A binary heatmap summarizing all identified virulence genes is shown in Appendix A, while Figure 3 shows a curated subset of key virulence factors. Based on the presence of pathotype-defining virulence genes, several isolates were classified as diarrheagenic *E. coli* (DEC): UNAP1–4 (Puno-Lampa) and UNAP7–10 (Puno-Melgar). UNAP6, although isolated from a healthy alpaca, also carried multiple virulence genes typically associated with DEC strains.

In a similar way to their resistance profiles, the virulence profiles of DEC isolates from Puno-Lampa and Puno-Melgar also generally followed their phylogenetic clustering, phylogroup classification, and strain characteristics. DEC isolates from Puno-Melgar (UNAP7,8,10, phylogroup B1, MLST 29, serotype O109:H11; UNAP9, phylogroup A, MLST 342, serotype O145:H25) harbored a complete Locus of enterocyte effacement (LEE) encoded Type III Secretion System (T3SS) along with additional virulence factors. In contrast, DEC isolates from Puno-Lampa (UNAP1–4; phylogroup B2, MLST 73, serotype O22:H1) lacked T3SS and instead carried a broader set of cytotoxins as well as invasion, immune evasion, iron acquisition, and adhesion factors. These observations show that the two DEC groups exhibit distinct virulence profiles, with LEE-encoded T3SS-dependent virulence in Puno-Melgar isolates and a T3SS-independent virulence strategy characterized by cytotoxins and other factors in Puno-Lampa isolates (Figure 3 and Appendix A).

UNAP7–10 carried additional virulence factors besides T3SS, including toxins (*hlyE* and *EAST1*, also known as *astA*) and immune evasion genes (*iss*, *ompT*, *katP*), though *ompT* and *katP* were absent in UNAP9. Some adherence factors (*paa*, *etgA*, *lpfA*, and *fdeC*) were primarily detected in UNAP7–10, though they were also found in other isolates. Furthermore, UNAP7–10, as well as UNAP6, harbored the Type III Secretion System (T3SS), including the master regulator *ler* and the *eae* gene, indicating the presence of the Locus of Enterocyte Effacement (LEE), a hallmark of enteropathogenic *E. coli* (EPEC). While *etgA* was detected in these genomes, no evidence of the EAF plasmid or *bfp* operon was found, supporting their classification as atypical EPEC (aEPEC), consistent with EnteroBase pathotype assignment (Figure 3).

Unlike UNAP7–10, UNAP1–4 lacked T3SS but, like nearly all isolates, carried genes encoding type I, type II, and type VI secretion systems (Appendix A). They harbored cytotoxins (*cnf1*, *vat*, *clbB*), hemolysins (*hlyA*), genes involved in mucin degradation and immune modulation (*pic*, *sslE*), bacteriocins and antimicrobial competition systems (*mchB*, *mchC*, *mchF*, *mcmA*, *cea*, *usp*), and the efflux pump gene *emrE*. Capsule-mediated immune evasion genes (*kpsMII*, *kpsE*), iron acquisition systems (*chuA*, *iroN*, *sitA*), and a distinct set of adhesion factors (*focACDG*, *sfaF*, *yfcV*, *hra*) were also detected (Figure 3). Given this virulence profile, which includes multiple ExPEC-associated virulence factors [14,27,28,29], UNAP1–4 are designated as ‘ExPEC-like’ isolates in this study.

Virulence gene content varied among the two isolates from healthy alpacas (UNAP5 and UNAP6, both from Puno-Lampa). UNAP5 (clustered with Puno-Melgar DEC isolates, in phylogroup B1) exhibited a smaller set of virulence genes, lacking T3SS and carrying fewer toxins (*subAB* as well as plasmid-encoded *cvaB* and *cvaC*), adherence factors, and immune evasion genes. In contrast, UNAP6 (clustered with UNAP9, a Puno-Melgar DEC isolate, in phylogroup A) carried numerous virulence factors, including the *eae* gene and an almost complete LEE-encoded T3SS, features consistent with its earlier classification as an atypical EPEC (aEPEC) (Figure 3).

Nearly all isolates presented iron acquisition systems (*entABCDEFS* and *fepABCDG* operons, *fes* gene), adherence factors (*fimH*, *csgAB*, *ompA*, and *yehABCD*), as well as stress response genes such as *ariR* (acid resistance and biofilm regulation) and *terC* (oxidative stress and tellurite resistance). Iron uptake genes associated with the yersiniabactin system (*fyuA*, *irp2*, *ybtP*, *ybtQ*) were also widely present.

### 3.7. Genomic Island and Plasmid Replicon Analysis

Multiple genomic islands were predicted across all *E. coli* genomes analyzed. Although most resistance determinants were chromosomal, fosfomycin resistance genes (*fosA7.5* and a *FosA* family gene) and efflux pump components (*emrY*, *emrK*, and *emrE*) were also found within predicted genomic islands (Appendix A). These regions also encompassed a substantial proportion of virulence factors, including ExPEC-associated genes in UNAP1–4 and LEE-encoded Type III Secretion System components in UNAP6–10 (Appendix A).

In addition to genomic islands, plasmids were also identified and analyzed. A total of 17 plasmid replicon types were detected across the 10 genomes, with individual isolates carrying between 1 and 5 replicons. UNAP10 harbored the highest number. The identified replicons included members of the IncF, IncY, ColRNAI, and Col156 groups (Table 2), with most sequences exhibiting high coverage (≥98%) and high nucleotide identity (≥94%) relative to reference replicon sequences (Appendix A). Plasmids did not carry resistance genes and encoded only a few virulence factors (Appendix A).

Plasmid analysis identified replicons such as IncFIB, IncFIC, and IncY, with varying coverage and identity percentages. These plasmids were associated with virulence factors and antibiotic resistance in some isolates. However, no direct antimicrobial resistance genes were detected on the plasmids, suggesting that resistance in these isolates was primarily chromosomal (Table 2 and Appendix A).

### 3.8. Pan-Genome Analysis of the 10 E. coli Isolates

The pan-genome analysis of the 10 *E. coli* isolated from sick and healthy alpacas in this study revealed a total of 6865 genes, with 3460 core genes (present in ≥99% of strains), 2025 shell genes (in 15–95% of strains) and 380 cloud genes (in less than 15% of the strains). The high proportion of accessory genes (shell and cloud genes) is indicative of high genomic plasticity among the strains. The pangenome heatmap (Figure 4) visually revealed distinct blocks of accessory genes present in DEC *E. coli* isolates from Puno-Lampa (Cluster 1, UNAP1–4) that were absent in those from Puno-Melgar (Cluster 2, UNAP7–10) and vice versa. These region-associated accessory gene sets aligned with previously identified differences in virulence gene content between sampling locations and were retained for downstream enrichment analyses (Figure 4).

Principal Component Analysis (PCA), applied to assess the impact of accessory genome variation, revealed three strain clusters that were broadly similar but not identical to those formed in the phylogenetic tree. Notably, UNAP5—although phylogenetically closer to UNAP7, UNAP8, and UNAP10—clustered instead with UNAP6 and UNAP9, despite their differences in phylogroup, region, and health status (Appendix A). Epidemiological factors such as geographic origin, health status, phylogroup, or MLST were not able to fully explain the observed clustering pattern (Appendix A). Nonetheless, a partial correspondence with geographic origin was observed (Appendix A)—particularly in the separation of DEC isolates from Puno-Lampa and Puno-Melgar—with the exception of UNAP5 and UNAP6 isolates from healthy-looking alpacas in Puno-Lampa. These two strains presented accessory genome content and virulence profiles that more closely resembled DEC isolates from Puno-Melgar than those from their region of origin, as visually confirmed by the pangenome heatmap (Figure 4).

### 3.9. Functional Categorization and Biological Pathway Analysis

GO enrichment of the core genome highlighted essential biological processes consistent with housekeeping functions (Appendix A). In contrast, enrichment of the accessory genome revealed (Appendix A) terms associated with response to toxic substances and virulence-related processes. To explore functional differences between the two DEC groups, GO, KEGG Orthology (KO), and KEGG Pathway enrichment analyses were performed on all genes uniquely found in either UNAP1–4 or UNAP7–10 (Figure 5, Figure 6 and Figure 7). These analyses identified functional enrichments consistent with the genomic differences already described between the groups.

Functional enrichment analysis of genes uniquely found in UNAP1–4 strains from Puno-Lampa confirmed a virulence and survival strategy distinct from that of UNAP7–10. These strains lack a Type III Secretion System (T3SS). Instead, they showed significant enrichment for alternative secretion mechanisms, including Type II (T2SS) (Figure 5), as well as Type I (T1SS) and Type VI Secretion Systems (T6SS) (Appendix A), supported by GO and KEGG pathway analysis. Together, these results suggest a T3SS-independent strategy for exporting virulence factors. Notably, enrichment of immune-related processes, including negative regulation of immune, leukocyte, and neutrophil activation (GO:0002683, GO:0002695, GO:1902564) (Figure 5), was linked to the *esiB* gene, suggesting mechanisms for immune evasion. Other enriched functions included colonization and persistence-related processes, such as biofilm formation—Vibrio cholerae (ko05111) (Figure 6) and iron uptake terms (Figure 7). Enrichment of the Pertussis pathway (ko05133) (Figure 7) suggests shared mechanisms with other pathogens, particularly involving adherence factors and type I hemolysin secretion (Appendix A).

In contrast, enrichment analysis of genes uniquely found in UNAP7–10 strains from Puno-Melgar revealed a virulence profile congruent with their classification as enteropathogenic *E. coli* (EPEC). Multiple terms and pathways indicate the presence of a Type III Secretion System (T3SS), including GO enrichment of Protein secretion by the type III secretion system (GO:0030254) (Figure 5), the KEGG Bacterial Secretion System (ko03070) pathway (Figure 6), and KO terms related to structural components and effectors (Figure 7). KEGG pathway enrichment also highlighted Bacterial invasion of epithelial cells (ko05100) and Pathogenic *E. coli* infection (ko05130) (Figure 6), confirming a T3SS-based strategy integrating structural proteins, effectors, and adhesins. Manipulation of host cytoskeletal architecture was reflected by enrichment of actin-associated terms such as actin cytoskeleton organization and polymerization or depolymerization (Figure 5), driven by a genomic island-encoded *TcdA/TcdB* catalytic glycosyltransferase domain-containing protein.

Other enriched host manipulation terms included symbiont-mediated killing of host cells, perturbation of host cell cycle progression, and intracellular transport (Figure 5). Finally, enrichment of an Escherichia phage integrase (ko:K21039) (Figure 7) indicates potential for horizontal gene transfer. Together, these enrichment results are consistent with a cytotoxic profile characteristic of diarrheagenic *E. coli* (DEC) strains of the EPEC pathotype.

Enrichment patterns based on group-specific virulence genes (Appendix A) were also explored, which further supported the main patterns observed. These enrichments provide a functional context for the genomic differences observed in each group and complement the virulence profiles previously described.

### 3.10. Comparative Genomics

A Minimum Spanning Tree (MST) based on cgMLST profiles was constructed to examine the genetic relatedness between the 10 *E. coli* isolates from alpacas sequenced in this study and 469 Peruvian *E. coli* genomes from different hosts available in EnteroBase (Figure 8). The alpaca isolates were assigned to distinct cgST Complexes and distributed across multiple MST clusters corresponding to different phylogroups, consistent with the previously observed presence of diverse lineages among the isolates from this study (Appendix A). UNAP1–4 (phylogroup B2, DEC) formed a tight subcluster within the cgST Complex 9 and were closely related to *E. coli* isolated from humans (Appendix A). Similarly, UNAP7, UNAP8, and UNAP10 (phylogroup B1, DEC and EPEC, Puno-Melgar) belonged to cgST Complex 2 and clustered with strains from humans (B1, EHEC) and swine (B1, EPEC), indicating that these isolates are part of a multi-host lineage (Appendix A). UNAP5 (B1, healthy alpaca isolate) also grouped near human isolates, while UNAP6 (A, EPEC) and UNAP9 (A, DEC and EPEC) clustered together with an environmental isolate from river water (phylogroup B1) rather than with other A phylogroup strains (Figure 8). Notably, the other four *E. coli* genomes isolated from alpacas available in EnteroBase (all phylogroup B1) in the MST were genetically distant from the isolates in this study, separated by multiple human-, swine-, poultry-, and water-associated strains, and did not form a single alpaca-associated genetic cluster (Figure 8).

Based on the MST, the closest genetic neighbors to the isolates from this study were selected for phylogenomic reconstruction (Appendix A). The phylogenomic tree revealed two main clades that largely followed the MST structure. UNAP1–4 (phylogroup B2) clustered tightly together in one clade, surrounded exclusively by *E. coli* strains from human origin (Figure 9). In the second major clade, isolates UNAP7, UNAP8, and UNAP10 (phylogroup B1, DEC, and EPEC) also formed a tight subcluster. This group was closely related to human-derived EPEC strains and swine-derived EPEC isolates (EC273K and EC275K) and shared a common ancestor with ECA1, an alpaca-derived EHEC strain (B1, Cusco), as well as a cluster of human EHEC isolates from Cusco (Figure 9).

UNAP6 and UNAP9 (phylogroup A, EPEC) clustered together in a distinct subclade with ECB1 (an alpaca-derived EPEC genome from Cusco) as the nearest outgroup. These isolates were more closely related to *E. coli* from environmental (river water) and human sources than to other phylogroup A strains, echoing their unusual placement in the MST (Figure 8). UNAP5 (B1, healthy alpaca) formed a separate subcluster within the B1-dominated portion of the tree. While it clustered near three other alpaca-derived genomes (PSU-0717, PSU-0719, PSU-0720), its closest neighbors were human-derived isolates from both Lima and Cusco (Figure 9).

## 4. Discussion

The findings of this study highlight the genetic diversity and pathogenic potential of *Escherichia coli* isolates from alpaca crias, shedding light on their role in both animal health and broader epidemiological contexts. The identification of predominant sequence types (STs) such as ST73 and ST29, alongside unique core genome sequence types (cgSTs), indicates a mixed population of *E. coli* in alpacas. This genetic variability is consistent with previous studies in livestock, where clonal complexes often reflect host-specific adaptation and environmental pressures [1,2].

The distribution of phylogroups (B2, B1, and A) revealed a clear association between phylogroup B2 and diarrheic alpacas, which aligns with its recognized pathogenicity in other hosts. Conversely, phylogroups A and B1 were predominant in healthy animals, suggesting a commensal role or reduced virulence potential. Serotyping further corroborated these findings, with pathogenic serotypes such as O22:H1 and O109:H11 enriched in diarrheic isolates. These results emphasize the need to monitor specific serotypes and phylogroups as potential indicators of disease outbreaks in alpacas [1,4].

The detection of antibiotic resistance-associated genes and mutations, including efflux pumps (*acrF*, *emrD*, *mdtM*), β-lactamase (*blaEC-15*), and fosfomycin resistance-related variants (*glpT_E448K*), suggests a concerning trend toward potential multidrug resistance (MDR) in *E. coli* isolates. The identification of colistin resistance-associated mutations (*pmrB_E123D* and *pmrB_Y358N*) is particularly notable, given the importance of colistin as a last-resort antibiotic in veterinary medicine. However, it is important to note that such genetic markers are not always predictive of phenotypic resistance, and further susceptibility testing is needed to confirm their functional impact [30]. These findings nonetheless underscore the urgency of implementing antimicrobial stewardship programs in alpaca farming to limit the spread of MDR *E. coli* [3,12].

Virulence gene profiling revealed diverse factors associated with adhesion (*fdeC*, *lpfA*), immune evasion (*iss*, *katP*), and toxin production (*hlyA*, *cnf1*). Pathotype analysis identified ExPEC-like strains in phylogroup B2, which are known for their systemic pathogenicity, and EPEC strains in phylogroup A, characterized by LEE pathogenicity island genes such as *eae* and *tir*. The coexistence of virulence and resistance genes in some isolates highlights the potential risk of these pathogens to both animal and public health [1,7].

The identification of resistance genes to biocides (*emrE*), acids (*ariR*), and metals (*terD*, *terW*, *terZ*) reflects the isolates’ ability to survive harsh environmental conditions. Plasmid analysis further revealed replicons associated with virulence and resistance, including IncFIB and IncFIC, which are known to play a role in horizontal gene transfer. However, the absence of direct antimicrobial resistance genes on these plasmids suggests that chromosomal mechanisms predominate in this *E. coli* population [5,6]. Although this analysis is based on a small number of isolates, the observed genetic patterns are consistent with previous studies reporting plasmid-encoded and chromosomally integrated resistance and virulence traits in *E. coli* from alpacas and other South American camelids [31,32,33].

The coexistence of pathogenic and antimicrobial resistance-associated genetic features in *E. coli* from alpacas raises concerns about zoonotic transmission and the potential for outbreaks in high-altitude farming regions. Although the presence of colistin resistance mutations is alarming due to the critical role of colistin in treating multidrug-resistant infections, these mutations should be interpreted with caution, as they do not always correlate with phenotypic resistance [30]. These findings highlight the need for routine monitoring and genomic surveillance of *E. coli* in alpaca populations, as well as the implementation of targeted management practices to reduce pathogen load and resistance spread [2,3]. Similar concerns have been raised globally, as alpacas and other South American camelids have been identified as potential reservoirs of *Shiga* toxin-producing and multidrug-resistant *E. coli*, with zoonotic implications for human and animal health [32,33,34].

The detection of *fimH* and *iss* genes in all *E. coli* strains isolated from compartment 1 of alpacas suggests their conserved role in colonization and potential virulence. *FimH*, which encodes the tip adhesin of type 1 fimbriae, is crucial for both mannose-specific and nonspecific adhesion and is often under positive selection due to its contribution to host and surface attachment [35]. Beyond adhesion, *FimH* can activate dendritic cells and stimulate Th1/Th17 immune responses, potentially exacerbating inflammation in compromised mucosa [36]. Similarly, *iss*, typically located on the ColV plasmid, enhances serum resistance by promoting group 4 capsule synthesis and reducing complement-mediated damage, even though chromosomal copies like bor are insufficient to confer this phenotype [37,38]. The ubiquitous presence of both genes, even in strains from clinically healthy animals, suggests they may represent core elements of *E. coli* adaptation and persistence in the host environment, warranting further evaluation of their expression and role in disease.

In this study, the *cnf1*, *clbB*, and *usp* genes were identified exclusively in *Escherichia coli* strains isolated from diseased alpacas in Lampa but were absent in strains from similarly affected animals in Melgar, suggesting potential geographic variation in virulence profiles. *cnf1* encodes a toxin that activates Rho-GTPases, promoting cytoskeletal rearrangement, disruption of cell junctions, and resistance to apoptosis, thereby contributing to chronic inflammation and enhancing intestinal colonization and tissue invasion [39,40]. *clbB*, part of the colibactin biosynthetic cluster regulated by ClbR, is involved in the production of this genotoxin and has been associated with phylogroup B2 strains, which are linked to greater clinical virulence [41,42]. The *usp* gene, encoded within the horizontally acquired PAI, is regulated by environmental cues through *TyrR* and *H-NS* and encodes a nuclease with genotoxic activity capable of inducing DNA damage and triggering the SOS response, although this effect may be counteracted by the protective function of *ClbS* [43,44]. The exclusive presence of these genes in strains from a single region may reflect localized selective pressures and niche-specific bacterial adaptation.

The detection of IncFIC(FII) and IncFIB(pB171) plasmids in genomes UNAP7, UNAP8, and UNAP10 suggests a strong association with antimicrobial resistance and virulence factors, features previously documented in both extraintestinal and enteropathogenic *E. coli* strains [45,46]. These replicons, typical of the IncF group, are known for disseminating genes such as *bla_CTX-M*, *iutA*, *iroN*, *iss*, and *hlyF*, which enhance bacterial survival under hostile conditions and contribute to systemic infection potential [47,48]. Additionally, the presence of IncY—a broad-host-range plasmid also linked to resistance to beta-lactams and colistin—reinforces the likelihood of horizontal gene transfer of resistance determinants in agricultural settings [49]. In contrast, the ColRNAI, Col156, and Col(MG828) replicons, although historically considered cryptic or of low clinical impact, have frequently been detected in pandemic lineages such as ST131, where they may act as satellite elements or encode bacteriocins that confer competitive advantages [47,48]. Altogether, this plasmid repertoire reveals a combination of mobile elements with pathogenic and adaptive potential, underscoring the need for genomic surveillance in animal reservoirs.

The pan-genome analysis of *E. coli* isolates from alpacas revealed substantial genomic plasticity, characterized by a large proportion of accessory genes, which are known to drive functional differentiation among strains [50,51]. The detection of distinct region-associated gene sets between DEC isolates from Puno-Lampa and Puno-Melgar underscores the role of the accessory genome in shaping localized pathogenic strategies. Specifically, the presence of accessory blocks correlated with either T3SS-independent or T3SS-dependent virulence profiles, suggesting differential acquisition of virulence factors and supporting the concept of pathotype diversification through horizontal gene transfer [52,53]. Moreover, the PCA clustering based on accessory gene content—independent of phylogeny, MLST, or health status—highlights that accessory genome variation may offer a more nuanced resolution of strain relationships than core genome phylogeny alone [53,54]. This phenomenon has been widely reported in *E. coli*, where accessory gene-driven differentiation underlies the emergence of hybrid or atypical strains with unexpected virulence potential [51,54]. Notably, the clustering of UNAP5 and UNAP6 with DEC strains from a different region suggests possible gene flow or shared selective pressures that could facilitate the emergence of virulent genotypes even in animals without clinical signs, emphasizing the epidemiological relevance of accessory genome surveillance in livestock pathogens.

The genomic characterization of the UNAP7–10 *E. coli* strains (isolated from alpacas in Puno-Melgar) reveals a virulence profile consistent with their classification as enteropathogenic *E. coli* (EPEC), marked by a complete Type III Secretion System (T3SS)—likely located within the LEE pathogenicity island—capable of injecting a broad repertoire of effectors (over 25) into the host cytosol [55,56]. The identified genes include structural components of the T3SS apparatus and adhesins such as intimin (encoded by the eae gene), in agreement with the ability of these strains to induce intimate adherence and form attaching and effacing (A/E) lesions characteristic of EPEC in the intestine [57,58]. Consistently, functional enrichment analysis showed significant overrepresentation of pathways related to T3SS and cytoskeletal remodeling, reflecting the activity of effectors that hijack the host actin machinery to generate microfilament “pedestals” and alter apical membrane architecture [57]. Beyond cytoskeletal manipulation, several injected effectors subvert host cellular responses: for instance, some inhibit phagocytosis (e.g., EspB, EspJ) or block key inflammatory pathways (NF-κB, NLRP3 inflammasome), contributing to both immune evasion and cytotoxic damage in the intestinal epithelium [57]. Notably, similar findings have been reported in EPEC strains of veterinary origin; for example, atypical isolates from cattle harbor analogous virulence gene repertoires and may even acquire *Shiga* toxin genes via bacteriophages, highlighting the genomic plasticity of pathogenic *E. coli* and its zoonotic potential [59]. In line with this, the detection of phage-type integrases in the UNAP7–10 strains suggests the presence of integrated pathogenicity islands acquired through horizontal gene transfer [60].

While this study provides a detailed genomic characterization of *E. coli* isolates from alpacas in the Peruvian highlands, the analysis was based on a targeted selection of representative strains. As such, the findings offer a foundational perspective rather than an exhaustive overview of the regional *E. coli* population structure. Broader epidemiological surveillance across more alpaca herds and time points will be valuable to further validate and expand upon the patterns observed.

Virulence gene profiling revealed a wide array of factors associated with adhesion, immune evasion, iron acquisition, and toxin production. Isolates from phylogroup B2 exhibited ExPEC-like characteristics with strong systemic pathogenicity potential. Enteropathogenic *E. coli* (EPEC) pathotypes were identified in phylogroup A isolates, characterized by the presence of LEE-encoded genes such as *eae*, *espA*, and *tir*. Virulence profiles provided insights into the epidemiological relevance of the isolates and their capacity to cause disease.

The clustering patterns observed in the MST and the phylogenomic tree suggest that alpaca colonization may involve multiple *E. coli* lineages rather than a single host-adapted group—a pattern consistent with findings in other animal hosts [61] and also previously reported in alpacas [62,63]. Several of the newly sequenced alpaca isolates formed cgST Complexes with strains from humans and swine, particularly EPEC and EHEC pathovars, highlighting the potential for cross-host or zoonotic transmission of *E. coli* lineages between humans and livestock [64,65,66]. These findings support the possibility that alpacas, like other livestock species, may act as reservoirs or intermediate hosts in the dissemination of pathogenic *E. coli* [67]. In addition to host-associated connections, some alpaca isolates shared phylogenetic proximity with environmental strains. UNAP6 and UNAP9 (phylogroup A), despite being from different regions, share similar virulence profiles and genomic islands with pathogenic potential. Notably, they were positioned near a river water isolate (phylogroup B1) in both the MST and phylogenomic tree. These observations raise the hypothesis that surface water may act as an environmental reservoir facilitating the acquisition or circulation of virulence genes in alpaca-associated *E. coli* via horizontal transfer of genomic islands or, alternatively, may reflect the distribution of a common strain across regions. This is consistent with previous reports describing surface water as a conduit for pathogenic *E. coli* strains and for the dissemination of virulence factors and ARGs via mobile genetic elements [68,69,70,71,72,73,74,75].

## 5. Conclusions

In conclusion, this study provides valuable insights into the genomic diversity, resistance mechanisms, and virulence potential of *E. coli* in alpacas. These findings underscore the importance of integrating genomic tools into veterinary diagnostics and public health strategies to address the challenges posed by emerging pathogens in livestock systems. These findings offer a valuable starting point for future large-scale genomic studies on *E. coli* in alpacas.

## Figures and Tables

**Figure 1 microorganisms-13-01533-f001:**
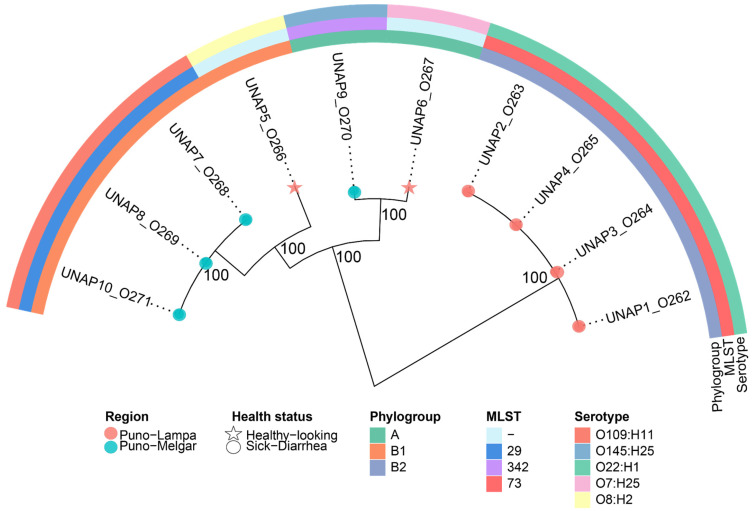
Maximum likelihood phylogenomic tree based on the core genome alignment of the ten *E. coli* strains isolated from diarrheic and healthy alpacas in Puno, Peru. The tree illustrates the evolutionary relationships among the isolates, with branch lengths representing genetic distances. Additional metadata, including Phylogroup, Region, MLST, Serotype, and Health status, is shown alongside the tree. Isolates from diarrheic and healthy alpacas are indicated with circles and stars at the tips, respectively.

**Figure 2 microorganisms-13-01533-f002:**
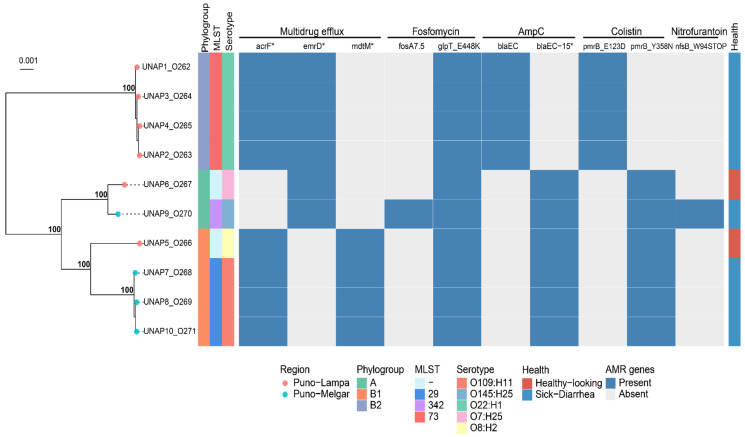
Binary heatmap indicating the presence/absence of ARGs across the 10 *E. coli* isolates. A phylogenomic tree of these strains is displayed on the left, annotated with Phylogroup, MLST, Serotype, and health status. Genes marked with an asterisk (*) represent putative variants or alleles identified through sequence similarity but not experimentally validated.

**Figure 3 microorganisms-13-01533-f003:**
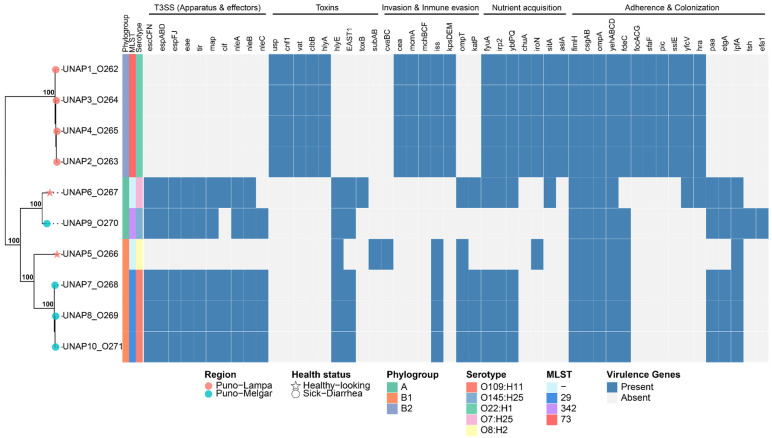
Binary heatmap showing the presence/absence of key virulence factors across the 10 *E. coli* isolates, based on a curated subset selected from the literature. A phylogenomic tree of these strains, annotated with phylogroup, MLST, serotype, and health status, is displayed on the left. Tree tips are marked with stars for isolates from healthy alpacas and with circles for DEC isolates.

**Figure 4 microorganisms-13-01533-f004:**
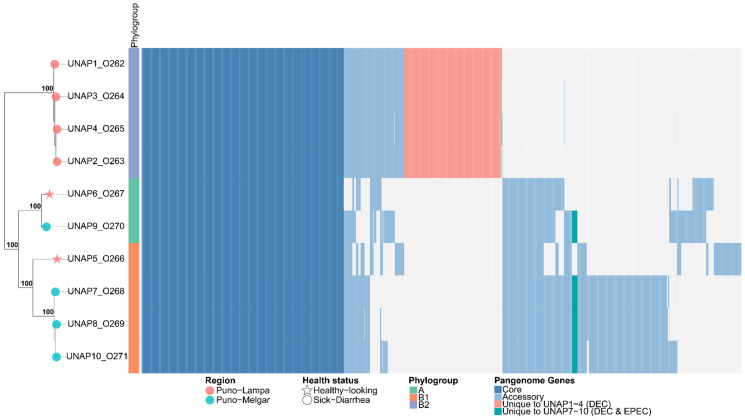
Heatmap of the pangenome composition of the *E. coli* isolates based on the gene presence/absence matrix and their core genome phylogeny. The phylogenetic tree (**left**) represents the evolutionary relationships among the ten *E. coli* genomes, while the heatmap (**right**) displays the presence (colored) and absence (white) of genes across the genomes. Core genes (≥99% of isolates) are shown in light blue, accessory genes (15–99% of isolates) in sky blue, and unique genes per genome (present in only one isolate) in red.

**Figure 5 microorganisms-13-01533-f005:**
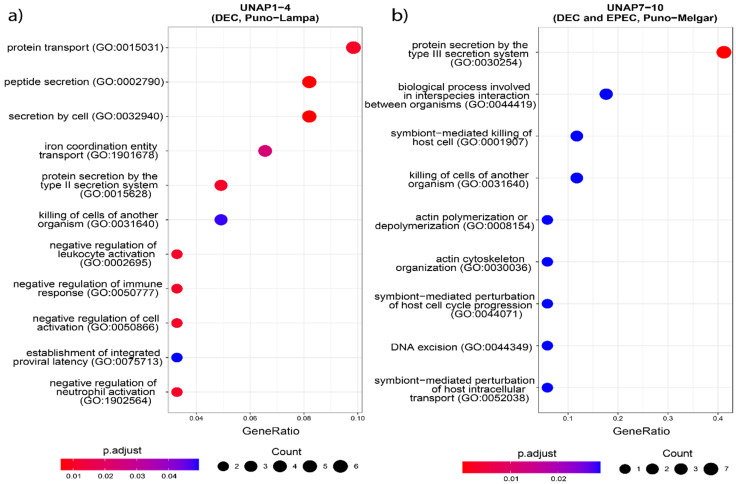
GO enrichment analysis based on the unique genes in both groups of DEC *E. coli* from the two sampling regions. Biological processes enriched in (**a**) UNAP1–4 (DEC from Puno-Lampa) and (**b**) UNAP7–10 (DEC and EPEC from Puno-Melgar).

**Figure 6 microorganisms-13-01533-f006:**
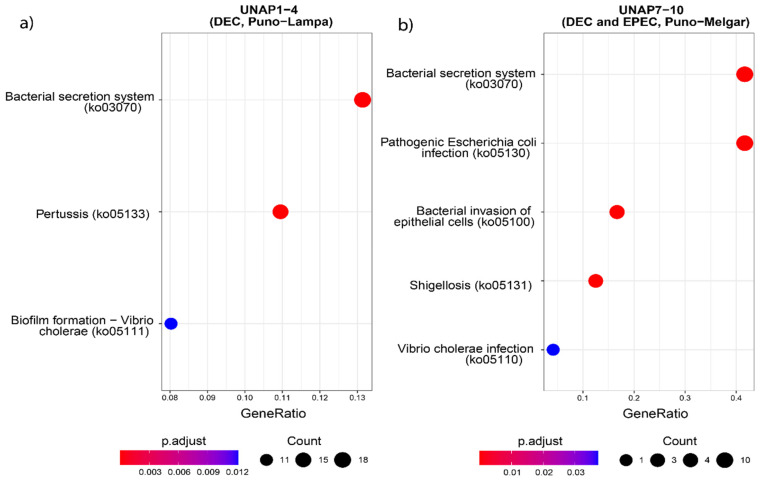
KEGG Pathway enrichment analysis based on the unique genes found in the groups of DEC *E. coli* from the two sampling regions. Pathways enriched in (**a**) UNAP1–4 (DEC from Puno-Lampa) and (**b**) UNAP7–10 (DEC and EPEC from Puno-Melgar).

**Figure 7 microorganisms-13-01533-f007:**
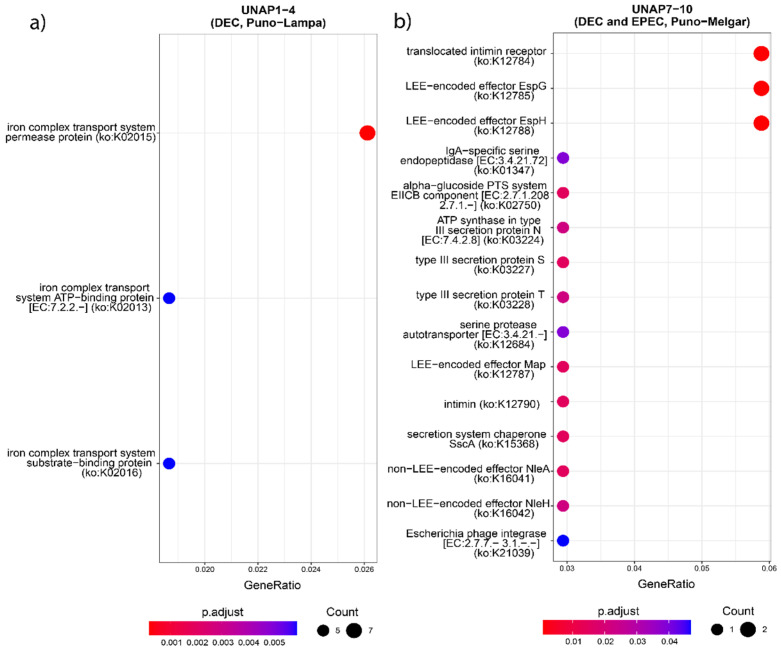
KEGG Orthology (KO) enrichment analysis based on the unique genes found in the groups of DEC *E. coli* from the two sampling regions. KO terms enriched in (**a**) UNAP1–4 (DEC from Puno-Lampa) and (**b**) UNAP7–10 (DEC and EPEC from Puno-Melgar).

**Figure 8 microorganisms-13-01533-f008:**
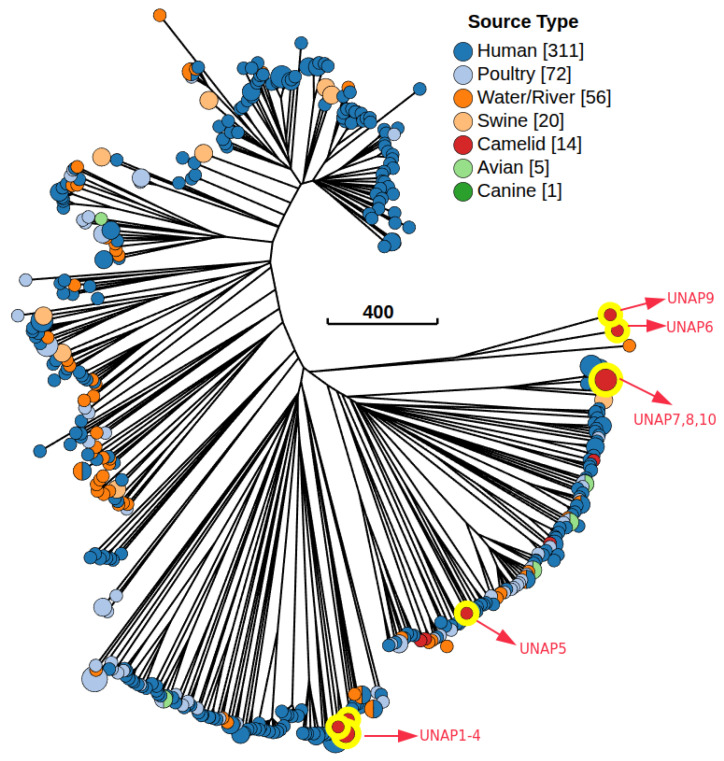
Minimum Spanning Tree (MST) based on the cgMLST V1 + Hierarchical Clustering (HierCC V1) scheme, including 479 *E. coli* genomes from Peru available in EnteroBase and the 10 isolates from alpacas for this study. The tree was constructed using the Neighbor-joining (RapidNJ) algorithm and visualized with GrapeTree in EnteroBase. Nodes are colored according to the isolation source. The red nodes highlighted in yellow are the *E. coli* genomes from Alpacas of the present study (10). The numbers in brackets are the number of isolates belonging to a particular source or host.

**Figure 9 microorganisms-13-01533-f009:**
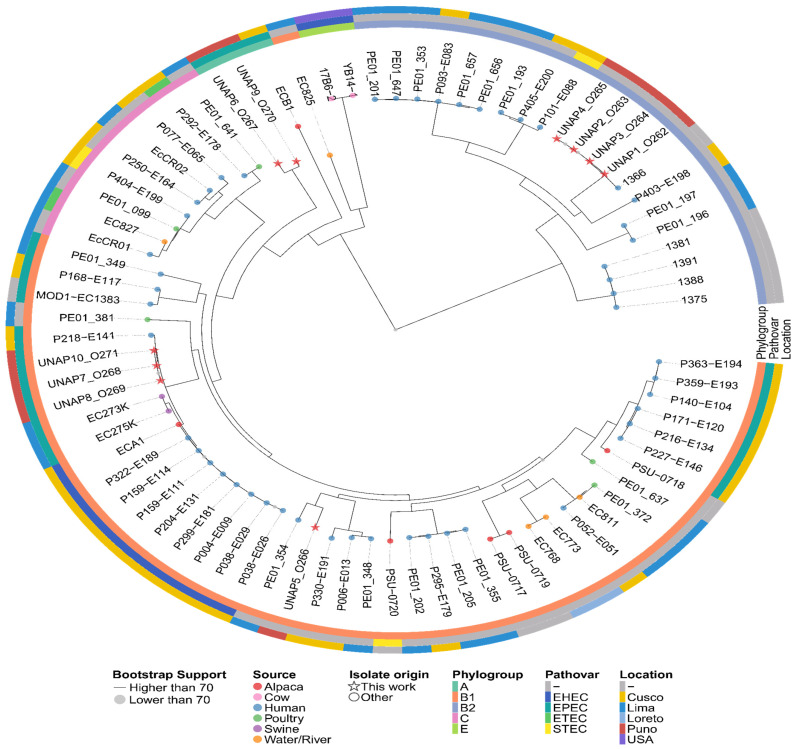
Maximum likelihood phylogenetic tree based on the core genome alignment of the 479 *E. coli* Peruvian genomes found in Enterobase and the 10 *E. coli* isolates of this study. The tree illustrates the evolutionary relationships among the isolates, with branch lengths representing genetic distances. Additional metadata, including Phylogroup, Pathovar, and Location, is shown alongside the tree. Isolates from this study are indicated with stars at the tips of the tree, while the rest are marked with circles. The color of the tips indicates the Host or Source of the *E. coli* isolates.

**Table 1 microorganisms-13-01533-t001:** Profile MLST, ST Complex, cgST, Phylogroup, and Serotype.

Isolate	Health Condition	Region	MLST	ST Complex	cgST	Phylogroup	Serotype
UNAP1_O262	Diarrhea	Puno	73	ST73 Cplx	321083	B2	O22:H1
UNAP2_O263	Diarrhea	Puno	73	ST73 Cplx	321084	B2	O22:H1
UNAP3_O264	Diarrhea	Puno	73	ST73 Cplx	321085	B2	O22:H1
UNAP4_O265	Diarrhea	Puno	73	ST73 Cplx	321084	B2	O22:H1
UNAP5_O266	Healthy	Puno	-	-	321086	B1	O8:H2
UNAP6_O267	Healthy	Puno	-	-	321088	A	O7:H25
UNAP7_O268	Diarrhea	Puno	29	ST29 Cplx	321087	B1	O109:H11
UNAP8_O269	Diarrhea	Puno	29	ST29 Cplx	321087	B1	O109:H11
UNAP9_O270	Diarrhea	Puno	342	-	321089	A	O145:H25
UNAP10_O271	Diarrhea	Puno	29	ST29 Cplx	321087	B1	O109:H11

**Table 2 microorganisms-13-01533-t002:** Summary of plasmids identified by Plasmifinder in *E. coli* isolated from alpacas.

Plasmid Marker	Description	Genomes
ColRNAI_1	A small plasmid with an RNA-based replication origin. Common in cloning vectors, not typically linked to resistance.	UNAP9_O270, UNAP10_O271
IncY_1	A broad-host-range plasmid often associated with antimicrobial resistance.	UNAP7_O268, UNAP8_O269, UNAP10_O271
IncFIC(FII)_1	Found in *E. coli*, often carrying multidrug resistance genes and virulence factors.	UNAP7_O268, UNAP8_O269, UNAP10_O271
IncFIB(pB171)_1_pB171	Associated with virulence and resistance in *E. coli*. Plays a role in pathogenesis and gene transfer.	UNAP7_O268, UNAP8_O269, UNAP10_O271
Col(MG828)_1	A small cryptic plasmid with no known association to resistance or virulence.	UNAP9_O270, UNAP10_O271
Col156_1	A small plasmid with limited host range, not strongly linked to resistance or virulence.	UNAP5_O266

## Data Availability

The original contributions presented in this study are included in the article/Appendix A. Further inquiries can be directed to the corresponding authors.

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
