# Peer review of "Genomic Characterization of Escherichia coli Isolates from Alpaca Crias (Vicugna pacos) in the Peruvian Highlands: Insights into Functional Diversity and Pathogenicity"

_microorganisms, 2025, doi:10.3390/microorganisms13071533_

Round 1
Reviewer 1 Report
Comments and Suggestions for Authors
The study by Zapa et al., deals with the genomic description of Escherichia coli isolates from Alpaca from the Puno region in Peru. The study is interesting to the reader and uses state of the art molecular analyses but the submitted manuscript needs improvement.
Comments to the authors:
The title and the introduction should clearly mention where the sampled Alpacas are from as today they can be found in many countries of the world.
The material and method section should include numbers of how many samples were investigated and how many isolates were further investigated by NGS. Furthermore, for cgMLST information is missing on how many and what samples were used for comparison of Alpaca isolates and what are the inclusion criteria.
Figure 1 is missing the red/yellow dots in the legend.
Supplementary table 1 should be included directly into the manuscript for better understanding. Also it´s just a small table.
In the results and discussion sections (138-143, 188-193) sentences using the word "predominantly" for the two isolates from the healthy animals are inapropriate as it is just one or two isolates. There is no true predominance here.
What ist the definition of ExPEC-like in this study?
The discussion on MDR in the analysed isolated is questionable (195-200, 217-219) as mentioned resistance markers are not always assocciated to the mentioned resistance/not 100% proof of phenotypical resistance (f.ex. https://pubmed.ncbi.nlm.nih.gov/39162501/).
Also generalizing the findings on the populations based on only 10 isolates should be put into the correct perspective. (213/214). Furthermore, the discussion is missing putting the results in the global context. There are already certain publication on Alpaca isolates/new world camelides existing (f.ex. PMID: 38200776, 29301877, 32990930, 32388703, 21493537, 15472347, 19407735).
The discussion and the conclusion is missing the limitations of the study.
Author Response
The study by Zapa et al., deals with the genomic description of Escherichia coli isolates from Alpaca from the Puno region in Peru. The study is interesting to the reader and uses state of the art molecular analyses but the submitted manuscript needs improvement.
-We sincerely thank the reviewer for their thoughtful and constructive evaluation of our manuscript. We appreciate your recognition of the relevance and interest of our study, as well as the use of advanced molecular analyses. We have carefully considered your comments and have made the necessary improvements to enhance the quality and clarity of the manuscript
Comments to the authors:
The title and the introduction should clearly mention where the sampled Alpacas are from as today they can be found in many countries of the world.
-Thank you for your valuable recommendation. In response, we have revised the title to:“Genomic Characterization of Escherichia coli Isolates From Alpaca Crias in the Peruvian Highlands: Insights Into Functional Diversity and Pathogenicity”
The material and method section should include numbers of how many samples were investigated and how many isolates were further investigated by NGS. Furthermore, for cgMLST information is missing on how many and what samples were used for comparison of Alpaca isolates and what are the inclusion criteria.
-Thank you for your observations. We edited to: “Fecal samples were obtained from eight alpaca crias showing symptoms of diarrhea, along with two samples collected from healthy individuals. All alpacas were maintained under the same living conditions (Table S1)”
Also, we appreciate your attention to the need for clarification regarding the cgMLST analysis. In response, we have added a detailed description to the Methods section:
“To investigate the evolutionary relationships and genomic diversity of the Escherichia coli isolates from alpacas, we conducted a phylogenomic analysis. For broader comparative context, we retrieved all 469 E. coli genomes from Peru available in EnteroBase (Dyer et al., 2025; Zhou et al., 2020, 2021), encompassing isolates from humans, swine, chickens, and alpacas. The 10 alpaca genomes were also uploaded to EnteroBase for joint clustering analysis. Core genome Multi-Locus Sequence Typing (cgMLST) was performed using the EnteroBase cgMLST V1 scheme, which profiles allelic variation across 2,513 conserved genes. Genomes were then grouped into hierarchical clusters (HierCC) based on cgMLST similarity. In E. coli, HierCC defines clusters at 11 levels based on cgMLST allelic distances; the HC1100 level corresponds to cgST Complexes, grouping genomes that differ by fewer than 1,100 core alleles (Zhou et al., 2021). A Minimum Spanning Tree (MST) was constructed using the RapidNJ algorithm and visualized in GrapeTree (Zhou et al., 2018) to explore genetic relatedness across isolates, and was colored by Host source, Phylogroup, Pathovars and HC1100 clusters representing cgST Complexes.
Based on the MST, 70 genomes representing the closest genetic neighbors to the alpaca isolates were selected for phylogenomic analysis. Selection was based on visual proximity to alpaca isolates in the MST, including those assigned to the same cgST Complexes (HC1100 level). Two additional genomes from alpacas in Peru and two from bovines in the USA were retrieved from NCBI to enhance representation of livestock-associated isolates. The selected genomes were processed with Panaroo 1.5.2 (Tonkin-Hill et al., 2020) to generate a core genome alignment, which was used to infer a maximum likelihood phylogenomic tree in IQ-TREE (Minh et al., 2020). ModelFinder (Kalyaanamoorthy et al., 2017) was used by IQ-TREE to identify the best-fit substitution model and branch support was assessed with 1,000 bootstrap replicates. The resulting tree was visualized and annotated with metadata such as host, geographic origin, and pathovar assignment using ggtree 3.14.0 (Yu, 2020)”.
Figure 1 is missing the red/yellow dots in the legend.
-Thank you for your observation. We added to the legend the next: “The red nodes highlighted in yellow are the E. coli genomes from Alpacas of the present study (10). The numbers in brackets are the number of isolates belonging to a particular source or host”.
Supplementary table 1 should be included directly into the manuscript for better understanding. Also it´s just a small table.
-Thank you for your recommendation. We edited table 1 with information from Supplementary table 1.
Table 1. Profile MLST, ST Complex, cgST, Phylogroup and Serotype
|
Isolate |
Health condition |
Region |
MLST |
ST Complex |
cgST |
Phylogroup |
Serotype |
|
UNAP1_O262 |
Diarrhea |
Puno |
73 |
ST73 Cplx |
321083 |
B2 |
O22:H1 |
|
UNAP2_O263 |
Diarrhea |
Puno |
73 |
ST73 Cplx |
321084 |
B2 |
O22:H1 |
|
UNAP3_O264 |
Diarrhea |
Puno |
73 |
ST73 Cplx |
321085 |
B2 |
O22:H1 |
|
UNAP4_O265 |
Diarrhea |
Puno |
73 |
ST73 Cplx |
321084 |
B2 |
O22:H1 |
|
UNAP5_O266 |
Healthy |
Puno |
- |
- |
321086 |
B1 |
O8:H2 |
|
UNAP6_O267 |
Healthy |
Puno |
- |
- |
321088 |
A |
O7:H25 |
|
UNAP7_O268 |
Diarrhea |
Puno |
29 |
ST29 Cplx |
321087 |
B1 |
O109:H11 |
|
UNAP8_O269 |
Diarrhea |
Puno |
29 |
ST29 Cplx |
321087 |
B1 |
O109:H11 |
|
UNAP9_O270 |
Diarrhea |
Puno |
342 |
- |
321089 |
A |
O145:H25 |
|
UNAP10_O271 |
Diarrhea |
Puno |
29 |
ST29 Cplx |
321087 |
B1 |
O109:H11 |
In the results and discussion sections (138-143, 188-193) sentences using the word "predominantly" for the two isolates from the healthy animals are inapropriate as it is just one or two isolates. There is no true predominance here.
Thank you for your observation. We deleted that word.
What ist the definition of ExPEC-like in this study?
Thank you for your observation. When we say ExPEC-like, it typically means the isolates carry virulence genes and features similar to ExPEC, but they have a slightly different genetic profile,
The discussion on MDR in the analysed isolated is questionable (195-200, 217-219) as mentioned resistance markers are not always assocciated to the mentioned resistance/not 100% proof of phenotypical resistance (f.ex. https://pubmed.ncbi.nlm.nih.gov/39162501/).
-Thank you for this insightful observation. We agree that the detection of resistance genes or point mutations does not provide definitive evidence of phenotypic resistance. Accordingly, we have revised the relevant sections of the Discussion to clarify that our conclusions are based on genotypic predictions, which should be interpreted with caution in the absence of phenotypic validation. The revised text reflects this limitation and includes reference to the need for further functional and phenotypic assessments to confirm antimicrobial resistance profiles.
Lines 195-200 The detection of antibiotic resistance-associated genes and mutations, including efflux pumps (acrF, emrD, mdtM), β-lactamase (blaEC-15), and fosfomycin resistance-related variants (glpT_E448K), suggests a concerning trend toward potential multidrug resistance (MDR) in E. coli isolates. The identification of colistin resistance-associated mutations (pmrB_E123D and pmrB_Y358N) is particularly notable, given the importance of colistin as a last-resort antibiotic in veterinary medicine. However, it is important to note that such genetic markers are not always predictive of phenotypic resistance, and further susceptibility testing is needed to confirm their functional impact (Butters et al., 2024). These findings nonetheless underscore the urgency of implementing antimicrobial stewardship programs in alpaca farming to limit the spread of MDR E. coli [3,11]
Line 217-219 The detection of antibiotic resistance-associated genes and mutations, including efflux pumps (acrF, emrD, mdtM), β-lactamase (blaEC-15), and fosfomycin resistance-related variants (glpT_E448K), suggests a concerning trend toward potential multidrug resistance (MDR) in E. coli isolates. The identification of colistin resistance-associated mutations (pmrB_E123D and pmrB_Y358N) is particularly notable, given the importance of colistin as a last-resort antibiotic in veterinary medicine. However, it is important to note that such genetic markers are not always predictive of phenotypic resistance, and further susceptibility testing is needed to confirm their functional impact (Butters et al., 2024). These findings nonetheless underscore the urgency of implementing antimicrobial stewardship programs in alpaca farming to limit the spread of MDR E. coli [3,11].
Also generalizing the findings on the populations based on only 10 isolates should be put into the correct perspective. (213/214). Furthermore, the discussion is missing putting the results in the global context. There are already certain publication on Alpaca isolates/new world camelides existing (f.ex. PMID: 38200776, 29301877, 32990930, 32388703, 21493537, 15472347, 19407735).
- Thank you very much for this valuable suggestion. In response, we have revised the Discussion to more clearly acknowledge the limitation of our small sample size and to avoid overgeneralization. Specifically, we now emphasize that the findings represent preliminary trends and are interpreted in the context of previous studies on E. coli from alpacas and South American camelids (see revised lines 213–214).
Additionally, we have incorporated several of the references provided (e.g., Aleman et al., 2024; Featherstone et al., 2011; Baranzoni et al., 2020; Mercado et al., 2004), which help place our results in a broader epidemiological and ecological context. This strengthens the global relevance of our findings and aligns them with current knowledge on E. coli virulence and resistance in camelid hosts.
Line 617-627 The identification of resistance genes to biocides (emrE), acids (ariR), and metals (terD, terW, terZ) reflects the isolates’ ability to survive harsh environmental conditions. Plasmid analysis further revealed replicons associated with virulence and resistance, including IncFIB and IncFIC, which are known to play a role in horizontal gene transfer. However, the absence of direct antimicrobial resistance genes on these plasmids suggests that chromosomal mechanisms predominate in this E. coli population [5,6]. Although this analysis is based on a small number of isolates, the observed genetic patterns are consistent with previous studies reporting plasmid-encoded and chromosomally integrated resistance and virulence traits in E. coli from alpacas and other South American camelids (Mercado et al., 2004; Featherstone et al., 2011; Maturrano et al., 2018).
Line 628-640 The coexistence of pathogenic and antimicrobial resistance-associated genetic features in E. coli from alpacas raises concerns about zoonotic transmission and the potential for outbreaks in high-altitude farming regions. Although the presence of colistin resistance mutations is alarming due to the critical role of colistin in treating multidrug-resistant infections, these mutations should be interpreted with caution, as they do not always correlate with phenotypic resistance (Butters et al., 2024). These findings highlight the need for routine monitoring and genomic surveillance of E. coli in alpaca populations, as well as the implementation of targeted management practices to reduce pathogen load and resistance spread [2,3]. Similar concerns have been raised globally, as alpacas and other South American camelids have been identified as potential reservoirs of Shiga toxin-producing and multidrug-resistant E. coli, with zoonotic implications for human and animal health (Alemán et al., 2024; Del Cacho et al., 2020; Baranzoni et al., 2020; Lee et al., 2011).
The discussion and the conclusion is missing the limitations of the study.
- Thank you for your valuable suggestion. We have now addressed this point by explicitly acknowledging the limitations of our study in the final paragraph of the Discussion section. We clarified that the findings are based on a representative subset of E. coli isolates from alpacas and emphasized the need for broader epidemiological surveillance to validate and expand these observations. Additionally, we added a brief statement in the Conclusion to highlight that this study serves as a starting point for future large-scale genomic investigations in alpacas.
Lines 721-726 While this study provides a detailed genomic characterization of E. coli isolates from alpacas in the Peruvian highlands, the analysis was based on a targeted selection of representative strains. As such, the findings offer a foundational perspective rather than an exhaustive overview of the regional E. coli population structure. Broader epidemiological surveillance across more alpaca herds and time points will be valuable to further validate and expand upon the patterns observed.
Lines 732-733 These findings offer a valuable starting point for future large-scale genomic studies on E. coli in alpacas.

Reviewer 2 Report
Comments and Suggestions for Authors
Overall, this is an interesting, concise, and clear manuscript. I have a few minor suggestions to improve the manuscript below:
1. E. coli genes need to be italicized (examples: abstract line 20, efflux genes on line 148, etc.)
2. In the introduction, you should include the proper Latin genus and species name of the alpaca (Llama pacos).
3. The Illumina sequence data should be provided as supplementary data.
4. Additional citations are needed in the discussion section. Notably, the second and third paragraphs of the discussion are under-cited. For example, authors need to cite how they concluded that phylogroup B2 correlates with pathological isolates from other animals. Colistin being used as a last resort antibiotic also should be cited.
Author Response
Reviewer 2
Overall, this is an interesting, concise, and clear manuscript. I have a few minor suggestions to improve the manuscript below:
- Thank you for your positive feedback and for recognizing the clarity and conciseness of our manuscript. We appreciate your careful review and have addressed your suggestions accordingly to improve the quality of the work.
- E. coli genes need to be italicized (examples: abstract line 20, efflux genes on line 148, etc.)
- Thank you for your observation. We have thoroughly reviewed the entire manuscript to ensure that all gene names, including E. coli genes (e.g., those mentioned in the abstract and on line 148), are properly italicized in accordance with standard scientific conventions.
- In the introduction, you should include the proper Latin genus and species name of the alpaca (Llama pacos).
- Thank you for your recommendation. We edited the title to: "Genomic Characterization of Escherichia coli Isolates From Alpaca Crias (Vicugna pacos) in the Peruvian Highlands: Insights Into Functional Diversity and Pathogenicity”
- The Illumina sequence data should be provided as supplementary data.
- Thank you for your comment. The next was added to the methods section: “The Illumina sequencing genomes have been deposited in the GenBank database of the NCBI and are publicly available under the following accession numbers: JBJFLZ000000000, JBJFMA000000000, JBJFMB000000000, JBJFMC000000000, JBJFMD000000000, JBJFME000000000, JBJFMF000000000, JBJFMG000000000, JBJFMH000000000, and JBJFMI000000000.”
- Additional citations are needed in the discussion section. Notably, the second and third paragraphs of the discussion are under-cited. For example, authors need to cite how they concluded that phylogroup B2 correlates with pathological isolates from other animals. Colistin being used as a last resort antibiotic also should be cited.
- Thank you for pointing this out. We agree that the statement regarding the association between phylogroup B2 and pathogenicity requires additional support. To address this, we have revised the discussion to include citations that demonstrate the pathogenic role of B2 strains specifically in animal hosts, as requested. We now cite Clermont et al., (2013) and Manges et al. (2019), which highlight the frequent association of B2 strains with extraintestinal pathogenic E. coli (ExPEC) infections. Kidsley et al. (2020) and Jakobsen et al. (2010) provide further evidence by showing that B2 strains isolated from dogs and meat-producing animals are capable of causing infections and exhibit virulence in animal models. Chakraborty et al. (2015) also notes that ExPEC strains from food-producing animals predominantly belong to phylogroup B2. These references have been incorporated into the revised discussion to support our conclusion that the B2 phylogroup is associated with pathogenicity in animal isolates. Additionally, we cite Nowrouzian et al. (2019) to illustrate that B2 strains include both uropathogenic and enteropathogenic lineages in the human gut. These references have been incorporated into the revised discussion paragraph to substantiate the statement about B2’s pathogenicity.
Finally, we have included citations to support the classification of colistin as a last-resort antibiotic, particularly for multidrug-resistant Gram-negative infections (Kaye et al., 2016; Mondal et al., 2024; Reardon, 2017). These references have been incorporated to substantiate the discussion on the clinical importance of colistin and the global concern regarding emerging resistance.
Clermont, O., Christenson, J. K., Denamur, E., & Gordon, D. M. (2013). The Clermont Escherichia coli phylo-typing method revisited: Improvement of specificity and detection of new phylo-groups. Environmental Microbiology Reports, 5(1), 58–65. https://doi.org/10.1111/1758-2229.12019
Manges, A. R., Geum, H. M., Guo, A., Edens, T. J., Fibke, C. D., & Pitout, J. D. D. (2019). Global Extraintestinal Pathogenic Escherichia coli (ExPEC) Lineages. Clinical Microbiology Reviews, 32(3), e00135-18. https://doi.org/10.1128/CMR.00135-18
Kidsley, A. K., O’Dea, M., Saputra, S., Jordan, D., Johnson, J. R., Gordon, D. M., Turni, C., Djordjevic, S. P., Abraham, S., & Trott, D. J. (2020). Genomic analysis of phylogenetic group B2 extraintestinal pathogenic E. coli causing infections in dogs in Australia. Veterinary Microbiology, 248, 108783. https://doi.org/10.1016/j.vetmic.2020.108783
Jakobsen, L., Hammerum, A. M., & Frimodt-Møller, N. (2010). Virulence of Escherichia coli B2 Isolates from Meat and Animals in a Murine Model of Ascending Urinary Tract Infection (UTI): Evidence that UTI Is a Zoonosis. Journal of Clinical Microbiology, 48(8), 2978–2980. https://doi.org/10.1128/JCM.00281-10
Chakraborty, A., Saralaya, V., Adhikari, P., Shenoy, S., Baliga, S., & Hegde, A. (2015). Characterization of Escherichia coli Phylogenetic Groups Associated with Extraintestinal Infections in South Indian Population. Annals of Medical and Health Sciences Research, 5(4), Article 4. https://www.ajol.info/index.php/amhsr/article/view/119720
Nowrouzian, F. L., Clermont, O., Edin, M., Östblom, A., Denamur, E., Wold, A. E., & Adlerberth, I. (2019). Escherichia coli B2 Phylogenetic Subgroups in the Infant Gut Microbiota: Predominance of Uropathogenic Lineages in Swedish Infants and Enteropathogenic Lineages in Pakistani Infants. Applied and Environmental Microbiology, 85(24), e01681-19. https://doi.org/10.1128/AEM.01681-19
Kaye, K. S., Pogue, J. M., Tran, T. B., Nation, R. L., & Li, J. (2016). Agents of Last Resort: Polymyxin Resistance. Infectious Disease Clinics, 30(2), 391–414. https://doi.org/10.1016/j.idc.2016.02.005
Mondal, A. H., Khare, K., Saxena, P., Debnath, P., Mukhopadhyay, K., & Yadav, D. (2024). A Review on Colistin Resistance: An Antibiotic of Last Resort. Microorganisms, 12(4), 772. https://doi.org/10.3390/microorganisms12040772
Reardon, S. (2017). Resistance to last-ditch antibiotic has spread farther than anticipated. Nature. https://doi.org/10.1038/nature.2017.22140

Round 2
Reviewer 1 Report
Comments and Suggestions for Authors
The authors have adequately responded to my comments in my review of their original manuscript.